# Understanding Callus Types in Maize by Genetic Mapping and Transcriptional Profiling

**DOI:** 10.3390/plants14203168

**Published:** 2025-10-15

**Authors:** Guifang Lin, Yan Liu, Tej Man Tamang, Yang Qin, Mingxia Zhao, Jun Zheng, Guoying Wang, Hairong Wei, Sunghun Park, Myeong-Je Cho, Frank F. White, Yunjun Liu, Sanzhen Liu

**Affiliations:** 1Department of Plant Pathology, Kansas State University, Manhattan, KS 66506, USA; guifanglin311@gmail.com (G.L.); ttamang@ycp.edu (T.M.T.); mingxia.zhao@pku-iaas.edu.cn (M.Z.); 2Basic Forestry and Proteomics Research Center, Fujian Agriculture and Forestry University, Fuzhou 350002, China; 3State Key Laboratory of Crop Gene Resources and Breeding, Institute of Crop Sciences, Chinese Academy of Agricultural Sciences, Beijing 100081, China; lyer09@163.com (Y.L.); 82101231090@caas.cn (Y.Q.); zhengjun02@caas.cn (J.Z.); wangguoying@caas.cn (G.W.); 4Environmental Horticulture, York College of Pennsylvania, York, PA 17403, USA; 5College of Forest Resources and Environmental Science, Michigan Technological University, Houghton, MI 49931, USA; hairong@mtu.edu; 6Department of Horticulture and Natural Resources, Kansas State University, Manhattan, KS 66506, USA; shpark@ksu.edu; 7Innovative Genomics Institute, University of California, Berkeley, CA 94704, USA; mjcho1223@berkeley.edu; 8Department of Plant Pathology, University of Florida, Gainesville, FL 32611, USA; ffwhite@ufl.edu

**Keywords:** regeneration, plant transformation, culturability, maize

## Abstract

Plant transformation efficiency is highly dependent on species, individual genotypes, and tissue types. In maize, immature embryos are regularly used for transformation. The process relies heavily on callus development, as it is intricately associated with somatic embryogenesis and subsequent plant regeneration, both of which directly affect transformation efficiency. Immature embryos of the segregation progeny derived from the two inbred parents, a transformation-amenable line A188 and a recalcitrant line B73, can be cultured to form two primary callus types: Type I and Type II. The Type II callus grows faster and is a favorable type for regeneration. Here, Type I and II calli from the B73xA188 F2 population were genotyped by Genotyping-By-Sequencing (GBS). Quantitative trait locus (QTL) analysis of the callus type identified QTLs at chromosomes 2, 5, 6, 8, and 9. The result was largely supported by the bulk segregant RNA-seq (BSR-seq) genetic analysis using RNA from separately pooled Type I and II calli. Both analyses revealed that an allele of A188 on chromosome 6 and B73 alleles on chromosomes 2, 5, 8, and 9 promoted the formation of the Type II callus. Differentially expressed genes (DEGs) between the Type II and I F2 calli were also identified. In addition, the A188 calli developed from the same immature embryos often exhibit heterogeneous morphology, including the fast- and slow-growing callus sectors. The transcriptional comparison between the two sectors was performed to identify DEGs. Both sets of DEGs were enriched in genes involved in cell-wall organization and wax biosynthesis pathways.

## 1. Introduction

Plant transformation is an essential technique for genome engineering, and efficient transformation protocols for diverse genetic backgrounds are desirable [1,2]. Maize transformation efficiency has been improved through genetic improvement [3], medium optimization [4,5,6,7], and morphogenic gene manipulation [8,9,10]. However, the transformation efficiency across cultivars or genotypes varies dramatically, and the underlying genetic basis remains largely unknown [1,2].

Immature embryos are often used as explants for DNA transformation in maize. Callus formation, somatic embryogenesis, and tissue regeneration are the major factors influencing the transformation efficiency [4,11,12]. Of 25 maize inbred lines surveyed, calli induced from immature embryos from A188, A634, W117, MS71, and H99 were highly regenerable, whereas the regeneration frequencies of calli from B73, H84, and N28 were low [12]. When crossed with A188, the regeneration efficiency of progeny of regeneration-recalcitrant lines B73, Mo17, H95, Oh43, and VA26 was markedly improved, indicating that A188 contains some dominant alleles for high regeneration ability [12].

Type I and Type II calli are two distinct embryogenic callus types that can be initiated from maize immature embryos. A Type I callus is translucent, typically slow growing, and of a compact structure mixed with differentiated tissue, while a Type II callus is white or pale yellow, fast growing and friable [11,13,14,15,16]. Most maize genotypes primarily produce a Type I callus [17]. While both Type I and II calli from different genotypes have been used to produce transgenic plants, the Type II callus is considered favorable for gene transformation due to its rapid growth and high regeneration efficiency [14,15,17,18,19].

Hi-II is a maize line with a high transformation efficiency [2,19,20]. Hi-II was generated from a cross of two partial inbred lines, Hi-II A and B, both of which exhibited nearly total Type II callus initiation from immature embryos [3,19,20]. Hi-II A and B were independently developed from crosses between B73, a transformation-recalcitrant elite breeding line, and A188, a highly regenerable line with poor agronomic qualities [12,21]. Type II calli occur upon culture of immature embryos from both A188 and B73 genotypes. However, the frequency of a Type II callus from B73 is very low [18].

The genetic basis of maize callus types is unknown. At the same time, high Type II callus initiation of Hi-II A and B and the benefit of crossing low regeneration lines with A188 support the possibility of callus regeneration improvement through genetic selection. Previously, a cross of B73xA188 was generated and backcrossed six generations to B73 (BC6), and individuals with the highest frequency of Type II callus were selected from each generation [22]. A single BC6 line was self-pollinated for four generations (BC6S4). The BC6S4 line carries five segments of A188 on chromosomes 1, 2, 3, and 9 on the basis of genotyping data of 73 RFLP markers. In a separate study, 89 genetic markers were used to identify culturability-associated regions on chromosomes 1, 3, 6, and 10 using progeny of Hi-II and a commercial inbred line FBLL [23]. A study of recombinant inbred lines of Mo17xH99, another low and high transformation efficient line cross, found QTLs related to callus formation at chromosomes 1, 2, 3, 5, 6, and 8 using 111 genetic markers [24]. A tissue culture response QTL between A188 and B73 was mapped to a region between around 164.8 Mb and 166.8 Mb on chromosome 3 (B73v3) [25]. A Wuschel-like homeobox 2a (*WOX2a*) gene enhancing maize regeneration was recently identified in the QTL region [26]. Overall, previous QTL studies were limited by the number of markers and segregating individuals.

Here, out of more than two thousand calli generated using immature embryos from self-pollinated B73xA188 F1 plants, distinct Type I and Type II F2 calli were collected for genetic mapping and transcriptomic profiling. Transcriptomic analysis was also performed on fast- and slow-growing callus sections of heterogeneous callus tissues of A188. Integration of genetic and transcriptomic data identified candidate genes that may be involved in callus development.

## 2. Results

### 2.1. Genetic Mapping of the Callus Type

Immature embryos (N = 2194) dissected from 13 self-pollinated B73xA188 F1 plants were cultured to select 100 XT-I and 100 XT-II calli [27]. One half of each individual XT-I or XT-II callus was subjected to Genotype-By-Sequencing (GBS). The other half of each callus was pooled into two XT-I and two XT-II bulks, with 50 calli per bulk for BSR-seq analysis.

GBS data were successfully generated for 153 individual calli, including 79 XT-I and 74 XT-II calli. Analysis of GBS data led to the identification of 96,703 SNPs. The SNP genotypes of each F2 individual callus were used to infer chromosomal segments harboring multiple SNP markers with the same genotypes (Appendix A). The number of segments per F2 individual represented the number of discernible recombination events. After filtering F2 individuals with recombination events higher than expected, 60 XT-I and 58 XT-II individuals were retained [28]. In total, 6369 GBS segment markers were generated for genetic mapping of the callus type (Figure 1A,B; Table 1). For the QTL mapping, a standard interval QTL mapping approach and a method of logistic regression were employed (Figure 1C, Appendix A). Concordant QTL peaks were identified by standard QTL and logistic regression on chromosomes 2, 5, 6, 8, and 9, which were designated as ctAB2a, ctAB5a, ctAB6a, ctAB8a, and ctAB9a, respectively (Table 1). The five QTLs explained 50.9% of the phenotypic variance. The LOD-supported QTL intervals were estimated, resulting in broad genomic ranges of the QTL intervals.

BSR-seq analysis of bulked XT-I and XT-II samples revealed 68,147 SNPs in the F2 population. Assuming read counts of two alleles of a SNP site followed a binomial distribution, the statistical test found 284 SNP sites associated with a divergent allele distribution in XT-I and XT-II pooled samples. Most associated SNPs were also located on chromosomes 2, 5, 6, 8, and 9 (Figure 1D; Table 1, Appendix A).

### 2.2. Alleles Contributing to Type II Callus Formation

Among the five QTLs, ctAB6a was the only QTL where the A188 genotype positively contributed to the Type II callus (Figure 2). QTL ctAB6a was located in a region of approximately 10 Mb to 100 Mb on chromosome 6 (Figure 2A,B). The distribution of three segregating genotypes showed that the A188 and B73 homozygous genotypes at the ctAB6a interval were enriched in XT-II and XT-I callus samples, respectively (Figure 2C,D). The heterozygous genotype was not strongly selected in either XT-I or XT-II, indicating non-dominant allelic impacts on callus types. QTL ctAB5a was the strongest QTL, which was mapped in the interval of 80 Mb to 198 Mb (Table 1, Figure 3A,B). The homozygous A188 and B73 genotypes of ctAB5a were enriched in XT-I and XT-II individuals, respectively, indicating that the B73 allele was favorable for the Type II callus (Figure 3C,D). Similar to QTL ctAB6a, the heterozygous genotype at QTL ctAB5a was not strongly selected in either XT-I or XT-II, indicative of non-dominant allelic impacts on callus types. In addition to ctAB5a, the B73 alleles were favorable alleles for Type II callus phenotype at the QTLs of ctAB2a, ctAB8a, and ctAB9a (Appendix A; Table 1). The phenotypic means of the three genotypes at all these four QTL peaks indicated that two alleles of all QTLs did not function in a dominant manner (Appendix A). Although we did not identify the chromosome 3 QTL that was previously mapped to condition tissue culture responses, we found that the B73 allele was under-represented in both XT-I and XT-II samples (Appendix A) [25]. A good proportion of individuals with the heterozygous genotype in both callus types indicated the impact from the A188 allele at the chromosome 3 locus on the tissue culture response was dominant (Appendix A).

### 2.3. Type II Callus Favorable Alleles of Many QTLs Were Selected in Both Hi-II A and B

Two highly culturable, nearly inbred lines selected from the progeny of A188 and B73, Hi-II A and B, were also subjected to GBS genotyping. Chromosomal segments of Hi-II A and B were inferred by GBS genotypes, showing that most chromosome regions were homozygous (Figure 4). Based on genotypes of segments, 22 and 54 recombination breakpoints were found in Hi-II A and B, respectively (Figure 4). Although Hi-II A and B were developed from independent F2 calli [3], the two lines shared genotypes of most chromosomal segments. The genotyping data showed that the Type II-callus-favorable B73 alleles at three QTLs, ctAB2a, ctAB6a, and ctAB8a, were selected in both Hi-II A and B. However, the Type II-callus-favorable B73 alleles at the major QTL, ctAB5a, and a minor QTL, ctAB9a, were not present in either line. Notably, both Hi-II A and B had the A188 genotype on the chromosome 3 QTL affecting tissue culture responses [25].

### 2.4. Differential Expression Between Different Types of Calli

Gene expression analysis between XT-II and XT-I calli identified 928 up-regulated and 753 down-regulated differentially expressed genes (DEGs) in XT-II as compared to XT-I (Appendix A). Enrichment analysis of GO terms in biological process and molecular function domains indicated that up-regulated genes were enriched in the pathways related to oxidant detoxification, stress response, biosynthesis of cell-wall components (cutin and wax), and heme binding, while the down-regulated genes were enriched in the pathways involved in transcription factor (TF) activity, transmembrane transport, protein dimerization, and nicotianamine metabolism (Figure 5A,B). In fast- and slow-growing calli, 935 up-regulated DEGs and 1509 down-regulated DEGs in fast-growing calli were identified in comparison to slow-growing calli (Appendix A). GO Enrichment analysis in the biological process and molecular function domains indicated that up-regulated DEGs were enriched in the process related to transcriptional regulation factors, transmembrane transport, wax and suberin biosynthesis, and hormone response. The down-regulated DEGs were enriched in the process of oxidative response, heme binding, response to oxidative stress, gibberellin 2-beta-dioxygenase activity, and cell-wall macromolecules (Figure 5C,D). The two differential expression analyses identified common TF DEGs. Eight TF genes were up-regulated in Type II and fast-growing calli as compared to Type I and slow-growing calli, respectively (Figure 5E). The up-regulated genes included two genes in the WOX family, Zm00001d039017 (*wox9c*) and Zm00001d043937 (*wox9b*), and two genes in the MYB family, Zm00001d043131 (*myb138*) and Zm00001d041853 (*myb8*). Note that both Zm00001d039017 and Zm00001d043937 were highly expressed in embryos, particularly in immature embryos, based on the publicly available expression atlas from MaizeGDB, while the two genes encoding MYB did not show this pattern (Appendix A) [29,30]. Conversely, 15 TF genes were down-regulated in Type II and fast-growing calli, including six genes from the ERF family and three genes from the bHLH family (Figure 5F).

Genes in biosynthesis of wax and fatty acid were enriched in the up-regulated DEGs. Detailed examination of the wax pathway found that five out of eight genes involved in the biosynthesis and transportation of cuticular wax were up-regulated in the Type II callus as compared to the Type I callus. These five genes include *gl2* (Zm00001d002353), *gl6* (Zm00001d041578), *gl8* (Zm00001d017111), *gl13* (Zm00001d039631), and *gl14* (Zm00001d004198) (Figure 6). Among these, the wax transmembrane transporter gene *gl13* [31] and a gene with unknown function *gl14* [32] were also up-regulated in the fast-growing calli as compared to the slow-growing calli.

### 2.5. Integration of Genetic Mapping and DEGs

The QTL intervals included 21 DEGs that were up-regulated in XT-II and fast-growing calli as compared with XT-I and slow-growing calli, respectively, and 28 DEGs that were down-regulated in XT-II and fast-growing calli (Appendix A). Of these 49 DEGs, six are in the interval of ctAB6a where the A188 allele type favored the formation of the Type II callus. One gene, Zm00001d036241 encoding GDSL esterase/lipase, was up-regulated, and five genes were down-regulated in both XT-II and fast-growing calli. Zm00001d036241 is presumably involved in lipid metabolism [34]. The five genes down-regulated in XT-II and fast-growing calli include *wat1* (walls are thin 1, Zm00001d036123) and *end1* (early nodulin homolog1, Zm00001d036125), which are homologs of genes involved in the nodule development, *si1* (silky1, Zm00001d036425) encoding a DNA binding protein relevant to the silk development, the gene Zm00001d036409 encoding an unknown function protein with C2-C2 zinc finger, and the gene Zm00001d035556 with an unknown function. Comparison of the A188 and B73 allele sequences found all these genes, except *si1*, exhibited nonsynonymous polymorphisms in exons.

The gene *wat1*, encoding an EamA-like transporter, was further examined. Comparison between the A188 and B73 allele sequences through Homotools [35] found two large insertions/deletions in the fourth intron and the 3′ untranslated region (3′ UTR), as well as three polymorphisms altering the protein product occurring in the fourth and fifth exons (Appendix A).

## 3. Discussion

Here, genomic loci associated with callus type were mapped to chromosomes 2, 5, 6, 8, and 9. Three of the QTLs mapped closely to previously reported QTL regions on chromosomes 2, 6, and 9 [2,3]. Alleles of the five QTLs favoring Type II calli were found in both A188 and B73 parents. The finding that the recombinant inbred lines (e.g., Hi-II A and B) developed a higher proportion of Type II calli than either parent during tissue culture indicated a combinational effect of the positive alleles. An allele of the QTL on chromosome 6 in A188 is associated with the Type II callus and may correspond with a previously reported regeneration-related QTL on chromosome 6 [23]. Our QTL analysis with GBS genotyping data of individual calli did not identify the chromosome 3 QTL or its associated *WOX2a* gene [25,26]. The failure to detect this QTL in our genetic analysis of F2 calli indicates that the chromosome 3 locus may not contribute to the determination of the callus type. Possibly, the QTL positively contributes to the regeneration ability of A188 and was not measured by our strategy. Although B73 does not carry this regeneration-favorable allele on chromosome 3, B73 has alleles favorable to Type II calli at multiple loci. In both Hi-II A and B lines, which form nearly 100% Type II calli and are highly regenerable, the chromosome 3 allele of A188 was selected with the addition of Type II-favorable alleles from A188 at ctAB06a and B73 at ctAB02a and ctAB08a. The combination of these alleles is, therefore, likely crucial for the regenerable lines from the progeny of A188 and B73.

Differential transcription comparisons between Type II and Type I calli, and between the fast- and slow-growing A188 calli, revealed that the genes in the pathway related to the cell-wall organization were enriched. One gene, *wat1* (Zm00001d036123), is located in the region of a QTL on chromosome 6. In *Arabidopsis*, *wat1* is involved in secondary cell-wall thickness [36]. Mutations in *Arabidopsis wat1* result in defective cell elongation due to abnormal secondary cell-wall formation in the fiber cells, leading to short stems. A *wat1* homologous gene was a susceptibility gene to a Gram-positive bacterium in tomato, which was thought to be functional through reducing auxin and ethylene content in plants [37]. The involvement of *wat1* in the production of phytohormones and their impacts on callus development needs further exploration. The cell wall plays an important role in plant development [38], and the DEGs other than *wat1* involved in cell-wall organization were identified here and in a previous transcriptomic study of callus induction in maize [39]. Down-regulation of genes in cell-wall pathways could lead to loosening of the cell wall and, possibly, cellular adhesion [38,40,41]. The difference in the cell-wall component and structure between Type I and II calli may result in variations in their ability to uptake water, oxygen, and nutrients, thereby affecting the growing rate and regeneration ability.

Transcriptomic comparisons between Type II and Type I calli and between the fast- and slow-growing A188 calli revealed an enrichment of TFs both as up-regulated and down-regulated DEGs. Up-regulated DEGs included members of the MYB and WOX families. The MYB TF Zm00001d043131, referred to as MYB138, was previously implicated in callus induction [42]. A T-DNA insertion mutant of the *myb138* homologous gene reduced callus formation as compared with the wildtype in *Arabidopsis*, presumably through enhancing gibberellin transduction, and promotes differentiation [42]. The *myb138* gene was up-regulated in fast-growing and Type II tissues, indicating that MYB138 may promote cellular proliferation and callus formation. The transcriptomic comparisons also identified two up-regulated genes encoding WOX TFs up-regulated in both fast-growing and Type II calli. In addition, one *ERF* gene was up-regulated, while six *ERF* genes were down-regulated in both fast-growing and Type II callus tissues. Multiple WOX and ERF TFs have been demonstrated to play critical roles in embryogenesis and stem cell maintenance in plants [43].

From the transcriptional analysis, genes in the fatty acid biosynthesis pathway are markedly enriched in the DEGs from the two RNA-seq analyses related to the callus type and callus growth. Very-long-chain fatty acids (VLCFAs), fatty acids with at least 20 carbons, are the precursors of many lipids, cuticular waxes, suberins, sphingolipids, and phospholipids [44]. In maize, a dozen genes, referred to as glossy genes, responsible for production or secretion of cuticular waxes have been cloned [32]. Although multiple glossy genes were up-regulated in Type II calli as compared to Type I calli and/or up-regulated in fast-growing calli as compared to slow-growing calli, no evidence supports that *gl4*, a carbon-condensing enzyme catalyzing a rate-limiting step in VLCFA biosynthesis [45], was up-regulated. Nevertheless, the expression elevation of the transmembrane transporter gene *gl13* [31] and other fatty acid modifiers indicates the alteration of intracellular VLCFA compositions. VLCFAs have been demonstrated to confine callus formation in *Arabidopsis* [46]. The reduction of VLCFAs could elevate the level of cytokinin, which fine-tunes cellular proliferation [47]. Further, altered VLCFAs may influence the production of ethylene [48]. Many ERF-encoding TF genes were down-regulated in the Type II and fast-growing callus tissues, which indicates the possibility of decreased levels of ethylene. In the future, monitoring amounts and compositions of VLCFAs in different types of callus tissues will provide better understanding of the roles of VLCFAs in the production of cytokinin and ethylene in the maize callus, and importantly, their impact on the signaling regulation of cell proliferation and regeneration.

Multiple genomic strategies have been employed to understand the genetic basis of the formation of the callus type in B73xA188. By combining genetic mapping and transcriptional analysis to gain deeper insights into the genetics underlying the formation of different callus types, we identified a set of DEGs, particularly those encoding TFs, with some located within broad QTL intervals, providing the candidate genes for functional validation. In the future, testing some DEGs through ectopic expression (e.g., *wox9b* and *wox9c*) or knockout (e.g., *wat1*) could further translate our findings for improving callus formation, somatic embryogenesis, and regeneration in tissue culture recalcitrant maize lines.

## 4. Materials and Methods

### 4.1. Establishment of B73xA188 F2 and A188 Calli

B73xA188 F_1_ plants were self-pollinated, and immature embryos (1.0–1.2 mm in length) were collected from 13 ears of 13 F2 plants at 11 days after pollination (DAP). Embryos were cultured in the dark at 28 °C for three weeks on N6 medium supplemented with 1.5 mg/L 2,4-dichlorophenoxyacetic acid (2,4-D) [49]. Out of 2194 total calli, 100 extremely Type I (XT-I) and 100 extremely Type II (XT-II) calli were selected. Each selected callus was bisected: one half was used for Genotyping-by-Sequencing (GBS), and the other for Bulked Segregant RNA-sequencing (BSR-seq). For BSR-seq, two biological replicates per callus type were generated, each consisting of 50 pooled calli, resulting in four total samples [50].

For the A188 line, 330 immature embryos were harvested from four ears at 11 DAP and cultured on N6 medium supplemented with 1.5 mg/L 2,4-D for 30 days, followed by subculture for 5 days. Type II calli from A188 often exhibited distinguishable fast- and slow-growing sectors upon closer inspection. These sectors were dissected from 20 representative calli and pooled into fast- and slow-growing bulks. Three biological replicates were collected for each callus growth type and subjected to RNA-seq.

### 4.2. DNA Isolation and GBS Sequencing

Leaves of ten-day seedlings from each maize line were collected for DNA extraction using the DNeasy Plant Mini Kit (Qiagen, Hilden, Germany). DNA samples were dissolved in water and normalized to 15 ng/μL for GBS library preparation.

A modified version of a previously published GBS protocol using the restriction enzyme of Bsp1286I (New England Biolabs, Ipswich, MA, USA) was employed for genotyping [21,51]. In brief, 150 ng genomic DNA of each individual sample was digested with Bsp1286I (New England Biolabs, Ipswich, MA, USA) at 37 °C for 2 h followed by the ligation of oligos as barcodes using T4 ligase (New England Biolabs, Ipswich, MA, USA) at 16 °C for 1.5 h. Both enzymes were inactivated at 65 °C for 20 min. Digested and ligated DNAs of numerous samples were pooled and purified together with a Qiaquick PCR purification kit (Qiagen, Hilden, Germany) and then AMPure XP beads (Beckman Coulter Life Sciences, Indianapolis, IN, USA). The purified products was subjected to PCR amplification with the Q5 high fidelity DNA polymerase (New England Biolabs, Ipswich, MA, USA) and the primers matching Illumina adaptors. PCR products purified using AMPure XP beads were sequenced on an Illumina HiseqX 10 platform at Novogene (Novogene, Sacramento, CA, USA), producing paired end 2 × 150 bp sequencing reads.

### 4.3. Genotypes of GBS Segment Markers

Illumina adaptors and low-quality sequences of raw reads were trimmed using Trimmomatic (version 0.38) [52], followed by decoding and fine trimming of customized adaptors with custom scripts to assign reads to each individual sample. To increase genotyping accuracy, whole genome sequencing (WGS) data of A188 (SRA accessions: SRR11870960 and SRR11870961) and B73 (SRA accessions: SRR4039069 and SRR4039070) were used to confirm genotypes of the two parental lines [21,53].

Reads were aligned to the B73 reference genome (B73Ref4) [54] with BWA [55]. Alignments were removed if they did not match any of the following criteria: 1. The insert size was in the range of 50–800 bp; 2. The mapping score was greater than 40; 3. The match region was greater than 50; 4. The mismatch percentage was less than 6%; and 5. The percentage of the unmatched overhang (or tail) was less than 5% of read length. SNPs were identified with GATK haplotypecaller [56,57], which were further filtered and converted to segment (bin) markers by using the R package (version 0.01) of Genomap (https://github.com/liu3zhenlab/genomap, accessed on 1 October 2021)

### 4.4. Genetic Mapping Using Logistic Regression

With GBS segment genotyping data of individual XT-I and XT-II samples, the logistic regression was employed to test for the null hypothesis that, at a marker site, genotype frequencies between the groups are not different. Significance thresholds were determined using two approaches: 1. Accounting for multiple tests with a false discovery rate (FDR) of 1% [58], and 2. Conducting 1000 permutation tests similar to a standard QTL permutation test to determine the distribution of *p*-values under the null hypothesis and selecting 5% as the significance *p*-value cutoff.

### 4.5. QTL Mapping Using R/qtl

Genetic positions of segment markers were estimated using a B73xA188 double haploid (DH) genetic map [21]. The R/qtl function scanone was used to map the QTLs with the standard interval mapping method and the binary model [59]. Two LOD thresholds were used: The LOD value at the 5% significance level from 1000 permutations and the LOD value of 3 [60]. The R/qtl function fitqtl was used to estimate the percentage of the QTL effect on the phenotype.

### 4.6. RNA Extraction and Sequencing of Bulked Samples

XT-I, XT-II, A188 fast- and slow-growing pooled tissue samples were ground with mortar and pestle under liquid nitrogen. RNA was isolated using the RNeasy Plant Mini Kit (Qiagen, Germantown, MD, USA) following the manufacturer’s instructions. Library preparation and RNA sequencing were performed at Novogene (Novogene, Beijing, China).

### 4.7. BSR-seq Analysis

RNA-seq raw reads were trimmed with Trimmomatic (version 0.38) [52] and aligned to B73Ref4 with STAR (version 2.7.3a) [61]. SNPs were called using GATK UnifiedGenotyper [56,57]. Bi-allelic SNPs were selected using GATK with the criteria “AF ≥ 0.2 && QUAL ≥ 30.0 && DP ≥ 100 && DP < 10,000”. Potential error SNPs that did not match polymorphic data between B73 and A188 were discarded. A generalized linear model assuming the binomial distribution of two alleles was employed to detect the association between each SNP and callus type.

### 4.8. Identification of Callus Type-Associated Genes

With read counts per gene resulting from STAR analysis, the statistical test with DESeq2 was performed to identify DEGs [62]. *p*-values were corrected to account for multiple tests with the FDR approach [58]. The genes with FDR-adjusted *p*-values smaller than 0.1 in the comparison were considered DEGs.

To identify candidate genes responsible for callus type, we focused on DEGs located within the QTL regions. Significant DEGs from the comparisons of XT-II versus XT-I and fast- versus slow-growing calli were prioritized as candidate genes.

### 4.9. Gene Ontology (GO) Enrichment Analysis

Gene Ontology (GO) enrichment analysis was performed using the resampling method in GOSeq [63]. Multiple testing adjustments were applied using the FDR method, with a significance threshold of FDR-adjusted *p*-values less than 0.05 [58].

### 4.10. Data Visualization

All figures, including QTL mapping and the genotyping segmentation of HI-II, XT-I and II samples, were plotted using custom R scripts. Heatmaps were plotted with the R package ComplexHeatmap [64]. The input data for heatmap plotting were standardized read counts per million of total reads (RPM). Standardization was achieved by subtracting the mean RPM across all samples and dividing the result by the standard deviation.

## 5. Conclusions

In this study, we employed two efficient genetic mapping strategies to dissect the genetic basis of callus type, Type I and Type II, derived from F2 immature embryos of the inbred lines B73 and A188, identifying five QTLs. Transcriptomic comparisons between morphologically contrasting calli revealed differentially expressed genes associated with callus development. By integrating genetic mapping with transcriptomic analysis, we identified candidate genes that may influence callus type, which can be further validated and applied to improve plant regeneration.

## Figures and Tables

**Figure 1 plants-14-03168-f001:**
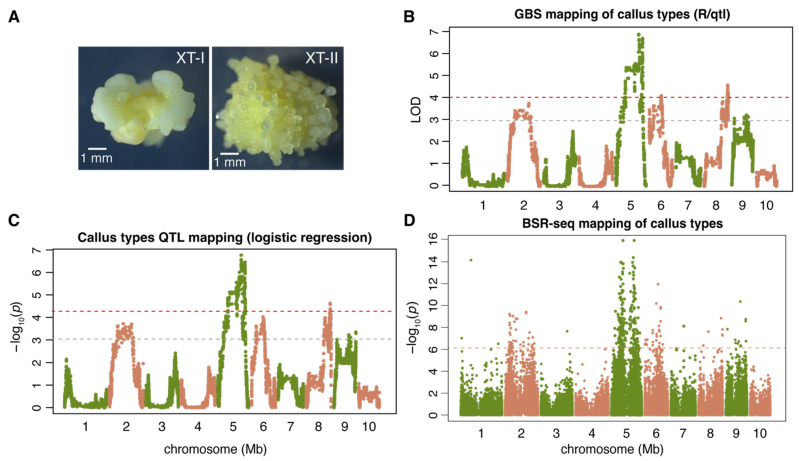
Genetic mapping of callus type QTLs. (**A**) Type I and II calli initiated from B73xA188 F2 I immature embryos. (**B**) Genetic mapping of the callus type using the standard interval mapping with a binary model. The red dash line indicates the significance threshold defined by the 1000 permutation tests at 5% significance level, and the gray dash line indicates the LOD threshold of 3. (**C**) The result of genetic mapping with a logistic regression method. The red dash line indicates the threshold defined by the 1000 permutation tests at the 5% significance level, and the gray dash line indicates the false discovery rate threshold of 0.01. (**D**) Genetic mapping of callus types via BSR-seq. The dash line indicates the threshold using the Bonferroni correction at the 5% significance level.

**Figure 2 plants-14-03168-f002:**
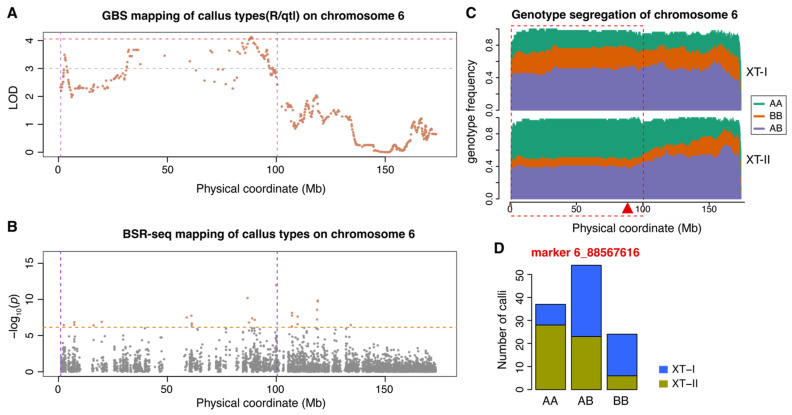
Detailed characterization of QTL ctAB6a. (**A**) Genetic mapping of callus types with GBS segment markers on chromosome 6. The red dash line indicates the significance threshold defined by permutation tests at the 5% significance level, and the gray dash line indicates the LOD threshold of 3. (**B**) Genetic mapping of callus types from BSR-seq. The orange dash line indicates the threshold defined by the Bonferroni correction at the 5% significance level. The significant SNP markers are colored in green. The vertical purple dash lines (in both (**A**) and (**B**)) indicate the LOD support QTL interval, and the red vertical dash line indicates the left flanking of the interval adjusted based on the BSR-seq mapping. (**C**) The distribution of genotypes of the 60 XT-I F2 calli and 58 XT-II F2 calli on chromosome 6. The red rectangle box indicates the QTL interval, and the red triangle points to the QTL peak. (**D**) Distribution of callus types in three genotypes. AA: A188 homozygous genotype; BB: B73 homozygous genotype; AB: heterozygous genotype.

**Figure 3 plants-14-03168-f003:**
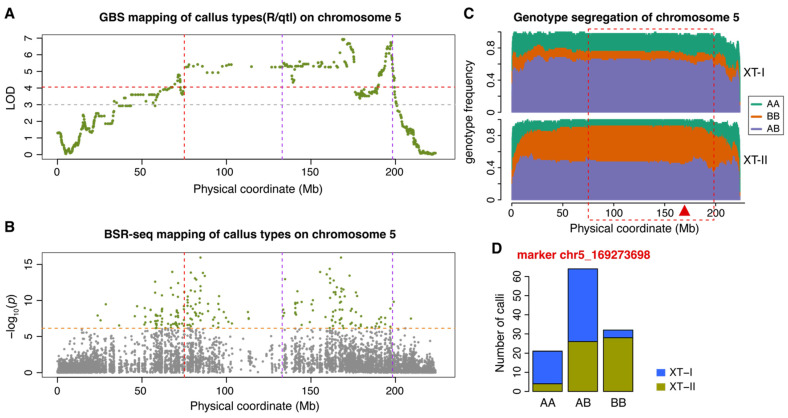
Detailed characterization of QTL ctAB5a. (**A**) Genetic mapping of callus types with GBS segment markers on chromosome 5. The red dash line indicates the significance threshold defined by permutation tests at the 5% significance level, and the gray dash line indicates the LOD threshold of 3. (**B**) Genetic mapping of callus types from BSR-seq. The orange dash line indicates the threshold defined by the Bonferroni correction at the 5% significance level. The significant SNP markers are colored in green. The vertical purple dash lines (in both (**A**) and (**B**)) indicate the LOD support QTL interval, and the red vertical dash line indicates the left flanking of the interval adjusted based on the BSR-seq mapping. (**C**) The distribution of genotypes of the 60 XT-I F2 calli and 58 XT-II F2 calli on chromosome 5. The red rectangle box indicates the QTL interval, and the red triangle points at the QTL peak. (**D**) Distribution of callus types in three genotypes. AA: A188 homozygous genotype; BB: B73 homozygous genotype; AB: heterozygous genotype.

**Figure 4 plants-14-03168-f004:**
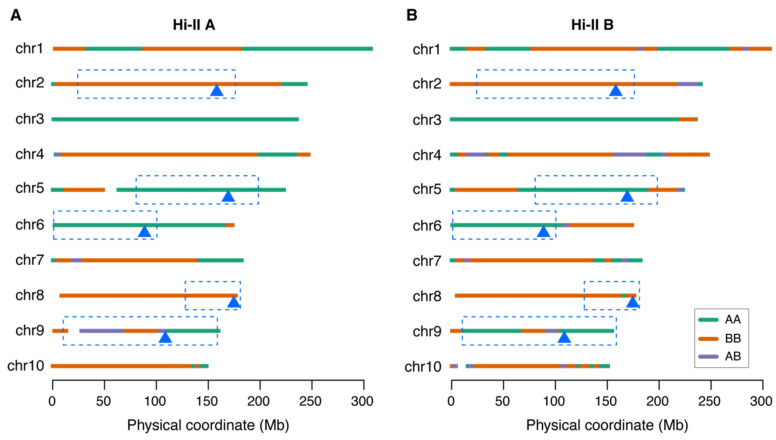
Genotypes of Hi-II A and B. (**A**,**B**) Genotype origins of chromosomal segments in Hi-II A (**A**) and B (**B**). The triangle under chromosomes labels the peak location of the QTLs, and the color of the triangle indicates the favorable allele. Green indicates the A188 allele, and orange indicates the B73 allele. Blue dotted rectangles indicate the QTL intervals. AA: A188 homozygous genotype; BB: B73 homozygous genotype; AB: heterozygous genotype.

**Figure 5 plants-14-03168-f005:**
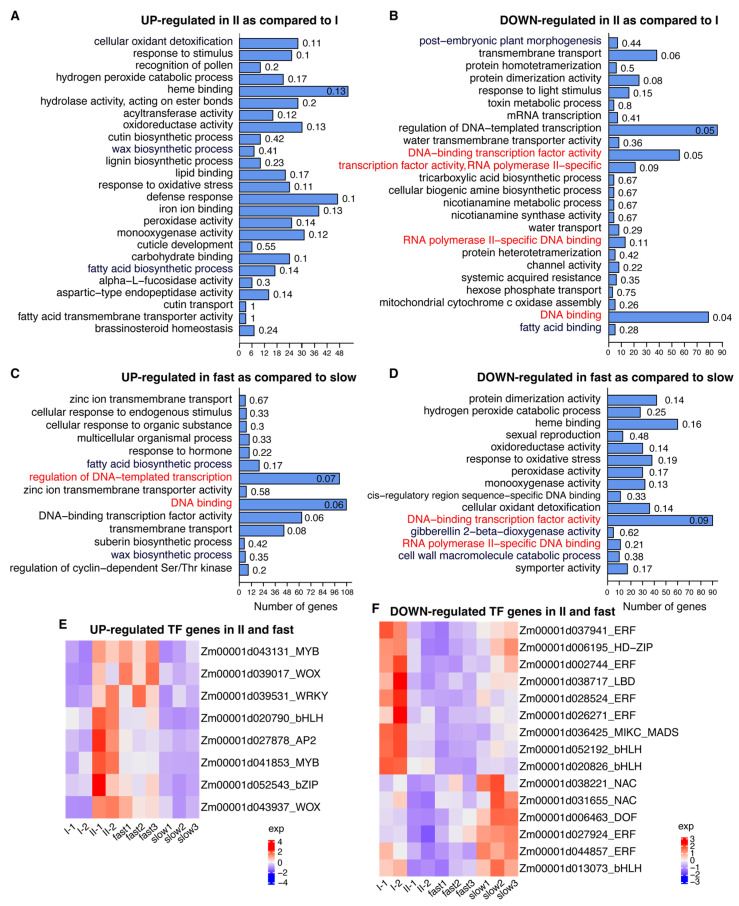
Enrichment of gene ontology (GO) in DEGs and expression of TF genes. (**A**,**B**) GO enrichment analysis of up- and down-regulated DEGs in XT-II (II) as compared to XT-I (I). (**C**,**D**) GO enrichment analysis of up- and down-regulated DEGs in fast-growing calli as compared to slow-growing calli. Each number on the top of a bar is the proportion of DEGs in the total genes of a certain GO. Red highlights GO terms related to transcription factor (TF) activities. (**E**) Heatmap of standardized expression of TFs that were up-regulated in XT-II as compared to XT-I and up-regulated in fast-growing calli as compared to slow-growing calli. (**F**) Heatmap of standardized expression of TFs that were down-regulated in XT-II as compared to XT-I and down-regulated in fast-growing calli as compared to slow-growing calli.

**Figure 6 plants-14-03168-f006:**
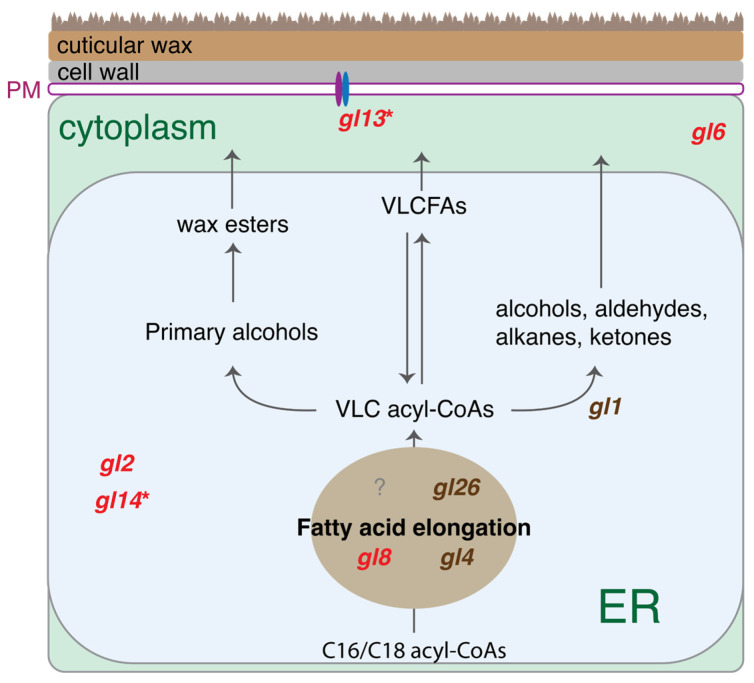
Genes in the pathway of cuticular wax production. The wax biosynthesis pathway was adapted from Figure 1B in Lewandowska et al. 2000 [33]. Very-long-chain fatty acids are produced and modified in the ER by various enzymes, resulting in wax components, which are then deposited to the plant surface through transmembrane transporters (e.g., GL13). Genes that are red-highlighted were up-regulated in the Type II callus as compared to the Type I callus. Two genes (*gl13* and *gl14*) labeled by an asterisk were up-regulated in the fast-growing callus as compared to the slow-growing callus. Genes that are brown-highlighted were not differentially expressed in both comparisons. VLCFA: very-long-chain fatty acid; VLC: very-long-chain; ER: endoplasmic reticulum; and PM: plasma membrane.

**Table 1 plants-14-03168-t001:** The QTLs supported by three mapping methods.

QTL	chr	R/qtl		Logistic Regression		BSR-seq		QTL Interval (bp) *	
		Pos (bp)	LOD	Pos (bp)	*p*-Value	Pos (bp)	*p*-Value	Start	End
ctAB5a	5	169,273,698	6.9	169,273,698	1.7 × 10^−7^	84,632,155	0.0 × 10^0^	80,458,296	198,270,432
ctAB8a	8	174,632,571	4.6	174,479,910	2.4 × 10^−5^	169,073,812	1.5 × 10^−9^	127,872,514	180,929,814
ctAB6a	6	88,567,616	4.1	83,296,319	9.6 × 10^−5^	99,960,681	1.1 × 10^−12^	1,006,642	100,419,946
ctAB2a	2	158,183,928	3.8	99,580,047	1.9 × 10^−4^	156,883,753	3.9 × 10^−10^	24,095,104	176,011,606
ctAB9a	9	108,566,128	3.3	159,752,864	4.5 × 10^−4^	108,540,812	4.4 × 10^−11^	10,213,832	158,723,248

* The interval was the LOD support interval, and CtAB.5.01 was adjusted to include the BSR-seq peak.

## Data Availability

Illumina sequencing data have been deposited in the Sequence Read Archive (SRA) database under accession PRJNA1192291. Scripts used in this study can be found at https://github.com/PlantG3/culturability.git (accessed on 1 December 2024).

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
