# Peer review of "Understanding Callus Types in Maize by Genetic Mapping and Transcriptional Profiling"

_plants, 2025, doi:10.3390/plants14203168_

Round 1

Reviewer 1 Report

Comments and Suggestions for Authors

Manuscript "Understanding callus types in maize by genetic mapping and transcriptional profiling" is very interesting.

General comments:
The authors investigated plant transformation efficiency as a function of genotypes and tissue types. The authors examined type I and II calli from the B73xA188 F2 population, which were genotyped by genotypic sequencing (GBS). The authors also performed a transcriptional comparison of the two sectors to identify DEGs.

Detailed comments:
The introduction is well-written, with references to recent world literature.
The citations are in a format that does not comply with the journal's requirements.
The description of the material is extensive and sufficient. Unfortunately, the description of the statistical methods is poor. The authors do not even mention the assumptions required for the application of some statistical methods, and testing them is crucial.
"log10" does not specify a base when writing the base-10 logarithm. This should be corrected.

Paper needs minor revision.

Author Response

The authors investigated plant transformation efficiency as a function of genotypes and tissue types. The authors examined type I and II calli from the B73xA188 F2 population, which were genotyped by genotypic sequencing (GBS). The authors also performed a transcriptional comparison of the two sectors to identify DEGs.

Comment 1: The introduction is well-written, with references to recent world literature.
The citations are in a format that does not comply with the journal's requirements.
The description of the material is extensive and sufficient. Unfortunately, the description of the statistical methods is poor. The authors do not even mention the assumptions required for the application of some statistical methods, and testing them is crucial.

Response 1: Thank you very much for positive comments and the suggestion to improve the description of the statistical hypothesis. For “Genetic Mapping Using Logistic Regression”, we have modified the hypothesis description: “With GBS segment genotyping data of individual XT-I and XT-II samples, the logistic regression was employed to test for the null hypothesis that, at a marker site, genotype frequencies between the groups are not different.”

Comment 2: "log10" does not specify a base when writing the base-10 logarithm. This should be corrected.

Response 2: The format of “log10” has been modified based on the suggestion. Thanks.

Reviewer 2 Report

Comments and Suggestions for Authors

The manuscript entitled “Understanding Callus Types in Maize by Genetic Mapping and Transcriptional Profiling” combines genetic mapping and transcriptional analysis to provide an in-depth study of the genetic basis underlying the formation of different callus types in maize.

Technically, the manuscript is well prepared, particularly regarding the QTL and BSR-seq analyses. The findings are valuable for improving callus regeneration in maize, as several candidate genes involved in callus development have been identified. According to the results, genes associated with cell wall organization are actively expressed in Type II calli and in fast-growing A188 calli.

I have one main concern. If other researchers wish to take advantage of the identified candidate genes, how long would it realistically take for them to improve gene transformation efficiency in their recalcitrant genotypes of interest? What steps would be required in this process? Providing some discussion on the translational potential and timeline of applying these findings would strengthen the manuscript.

Finally, I noticed that the manuscript lacks a Conclusion section. Adding a concise conclusion that highlights the key findings and their potential applications would improve the overall readability and impact of the paper.

Minor comments:

Abstract:

  • Line 25: “Of the segregation progeny …”. This sentence is unclear. Please re-write your paragraph.
  • Lines 67-69: We know that it is difficult to transfer a gene to tissues with rapid growth, like meristem nodes. So, how you interpret your mentioned sentences here?

Materials and Methods:

  • - Lines 190-191: Why you only focused on DEGs located within the QTL region on chromosome 6?
  • Lines 194-197: Typing font is changed in this paragraph.

Results:

  • Lines 207-210: This paragraph is a part of M&M not Results.
  • Lines 244-245: I can’t find these results in Table 1.

Author Response

The manuscript entitled “Understanding Callus Types in Maize by Genetic Mapping and Transcriptional Profiling” combines genetic mapping and transcriptional analysis to provide an in-depth study of the genetic basis underlying the formation of different callus types in maize.

Technically, the manuscript is well prepared, particularly regarding the QTL and BSR-seq analyses. The findings are valuable for improving callus regeneration in maize, as several candidate genes involved in callus development have been identified. According to the results, genes associated with cell wall organization are actively expressed in Type II calli and in fast-growing A188 calli.

Comment 1: I have one main concern. If other researchers wish to take advantage of the identified candidate genes, how long would it realistically take for them to improve gene transformation efficiency in their recalcitrant genotypes of interest? What steps would be required in this process? Providing some discussion on the translational potential and timeline of applying these findings would strengthen the manuscript.

Response 1: Thank you for the suggestion. To discuss the timeline for applying our findings is challenging. However, the paths to translate our findings are relatively clear. We would suggest conducting ectopic expression of a few candidate genes in recalcitrant lines, or knocking out some other genes for their impacts on callus development and regeneration. We have included the following statement in the Discussion:

“By combining genetic mapping and transcriptional analysis to gain deeper insights into the genetics underlying the formation of different callus types, we identified a set of DEGs, particularly those encoding TFs, with some located within broad QTLs intervals, providing the candidate genes for functional validation. In the future, testing some DEGs through ectopic expression (e.g., wox9b and wox9c) or knockout (e.g., wat1) could further translate our findings for improving callus formation, somatic embryogenesis, and regeneration in tissue culture recalcitrant maize lines.”

Comment 2: Finally, I noticed that the manuscript lacks a Conclusion section. Adding a concise conclusion that highlights the key findings and their potential applications would improve the overall readability and impact of the paper.

Response 2: We have added a conclusion at the end of the manuscript. Here is the paragraph:

“In this study, we employed two efficient genetic mapping strategies to dissect the genetic basis of callus type, Type I and Type II, derived from Fâ‚‚ immature embryos of the inbred lines B73 and A188, identifying five QTLs. Transcriptomic comparisons between mor-phologically contrasting calli revealed differentially expressed genes associated with callus development. By integrating genetic mapping with transcriptomic analysis, we identified candidate genes that may influence callus type, which can be further validated and applied to improve plant regeneration.”

Comment 3: Line 25: “Of the segregation progeny …”. This sentence is unclear. Please re-write your paragraph.

Response 3: The sentence has been revised. It now reads, “Immature embryos of the segregation progeny derived from the two inbred parents, a transformation-amenable line A188 and a recalcitrant line B73, can be cultured to form two primary callus types: Type I and Type II.”.

Comment 4: Lines 67-69: We know that it is difficult to transfer a gene to tissues with rapid growth, like meristem nodes. So, how you interpret your mentioned sentences here?

Response 4: DNA transformation in maize typically is conducted before callus is induced. No data so far indicate that immature embryos forming Type II callus are harder to transform.

Comment 5: Lines 190-191: Why you only focused on DEGs located within the QTL region on chromosome 6?

Response 5: Thank the reviewer very much for catching this. We have removed “on chromosome 6”.

Comment 6: Lines 194-197: Typing font is changed in this paragraph.

Response 6: this has been fixed.

Comment 7: Lines 207-210: This paragraph is a part of M&M not Results.

Response 7: We thank the reviewer for the suggestion. This paragraph introduces the experimental design. The number of immature embryos and the number of calli are part of the results. We think the paragraph would be better to stay in the Results.

Comment 8: Lines 244-245: I can’t find these results in Table 1.

Response 8: The reviewer is correct. We have changed the reference from Table 1 to Figure 2. Thanks.

Reviewer 3 Report

Comments and Suggestions for Authors

Plant transformation efficiency in maize depends on genotype and callus type, with Type II callus being more favorable for regeneration. QTL and BSR-seq analyses revealed that A188 alleles on chromosome 6 and B73 alleles on chromosomes 2, 5, 8, and 9 promote Type II callus formation. Differentially expressed genes between callus types and growth sectors were enriched in cell wall organization and wax biosynthesis pathways.

Belo are the comments;

Lines 49–51: You mention that the genetic basis of transformation efficiency remains largely unknown. Given multiple QTL and transcriptomic studies cited later, how does this work go beyond prior findings to provide novel insights?

Lines 55–59: The authors report improved regeneration efficiency in progeny when crossing B73 with A188. Can the authors clarify whether the observed improvement is due to specific dominant alleles from A188, or could maternal effects, epigenetic regulation, or culture conditions also play a role?

Lines 79–87: Previous QTL studies identified culturability-associated regions on several chromosomes. How do the authors plan to address the limitations of earlier studies (e.g., low marker density, small population sizes) in their own design?

Lines 207–211: The authors selected only 100 XT-I and 100 XT-II calli from over 2,000 embryos. How was this sample size determined, and could the relatively small subset bias the mapping results?

Lines 239–241: The BSR-seq analysis found 284 SNPs associated with callus type. How many of these overlap with the GBS-identified SNPs, and could the authors quantify the degree of concordance to validate robustness?

Lines 250–256: For ctAB5a, the B73 allele is favorable for Type II callus, yet A188 is typically described as the high-regeneration line. How do the authors reconcile this apparent contradiction?

Lines 415–417: The link between cell wall differences and nutrient/water/oxygen uptake is speculative. Do the authors have direct measurements (e.g., metabolite assays, ion flux studies) to support this, or is it entirely inferred from DEG functions?

Lines 421–425: The up-regulation of MYB138 is interpreted as promoting callus proliferation, but in Arabidopsis, it reduces callus formation via enhanced GA signaling. Could the opposite correlation in maize reflect species-specific roles, or might it suggest misinterpretation?

Lines 426–431: WOX and ERF transcription factors are discussed as regulators of embryogenesis. Were any of these WOX/ERF DEGs located within the mapped QTL intervals, which would strengthen their candidacy?

Lines 448–451: The authors recommend monitoring VLCFA levels in future work. Why was lipidomic profiling not included in this study, given that transcriptomic changes strongly pointed toward fatty acid metabolism?

Author Response

Plant transformation efficiency in maize depends on genotype and callus type, with Type II callus being more favorable for regeneration. QTL and BSR-seq analyses revealed that A188 alleles on chromosome 6 and B73 alleles on chromosomes 2, 5, 8, and 9 promote Type II callus formation. Differentially expressed genes between callus types and growth sectors were enriched in cell wall organization and wax biosynthesis pathways. Below are the comments;

Comment 1: Lines 49–51: You mention that the genetic basis of transformation efficiency remains largely unknown. Given multiple QTL and transcriptomic studies cited later, how does this work go beyond prior findings to provide novel insights?

Response 1: We employed new analyzing strategies to explore the genetic basis of callus development. Compared to previously studies, our QTLs are more relevant to callus types, and high-density markers were used. In addition, our data provide insights about transcriptomic changes between morphologically distinct calli. Specifically, we identified multiple genes encoding transcription factors that are potentially associated with callus development. The genes involved in cell wall formation and cuticular wax pathways were found to be associated as well. The genes mentioned above are novel genes related to callus development and a few of them are in the QTL intervals, such as the gene wat1.

Comment 2: Lines 55–59: The authors report improved regeneration efficiency in progeny when crossing B73 with A188. Can the authors clarify whether the observed improvement is due to specific dominant alleles from A188, or could maternal effects, epigenetic regulation, or culture conditions also play a role?

Response 2: Thank the reviewer for this important question. Our manuscript was designed based on previous data showing that B73 and A188 are regeneration-recalcitrant and -amendable lines, respectively, and that their progeny segregate in culturability and regeneration ability. It is no doubt that the culture condition is a very important factor for tissue culture. The experiments comparing the culturability of different genotypes were based on measurements on the same culture conditions. To date, the impacts from maternal effects and epigenetic regulation remain unclear in maize. Genetics studies clearly indicated that the ability of regeneration is controlled by multiple QTLs. Our manuscript is focused on the genetic basis of the development of callus types, which indicate the culturability and regeneration ability. Our data did not clearly evidence the dominant effect from A188 alleles of any of callus type QTLs. However, our data indicate that the impact from the A188 allele at the chromosome 3 locus on the tissue culture response was dominant. We have added this implication to the manuscript.

Comment 3: Lines 79–87: Previous QTL studies identified culturability-associated regions on several chromosomes. How do the authors plan to address the limitations of earlier studies (e.g., low marker density, small population sizes) in their own design?

Response 3: In the Introduction, we mentioned that previous QTL studies were limited by the number of markers and segregating individuals. For example, in Lowe et al 2026 study, only 89 genetic markers were used to identify culturability-associated regions. First, with a more complete maize reference genome and improved sequencing technologies, genetic makers are easier to generated. In both BSA-seq and GBS mapping, marker number does not pose a constraint. We have sufficient markers of our mapping purpose. Second, we employed an efficient strategy to study a difficult trait such as callus types. Traditionally, investigators might develop a mapping population (e.g., recombinant inbred lines or double haploids) followed by genotyping each line and phenotyping multiple individuals per line. In each individual, many immature embryos need to be examined to have reliable phenotypic data. In our method, we only grew 13 F1 plants and >2,000 individual F2 immature embryos were examined. This is very labor-efficient. In addition, we only sampled the calli with clear phenotype, which reduced phenotyping variation. In summary, our study deploys a new way to efficiently examine this complicated callus trait.

Comment 4: Lines 207–211: The authors selected only 100 XT-I and 100 XT-II calli from over 2,000 embryos. How was this sample size determined, and could the relatively small subset bias the mapping results?

Response 4: In our previously study that employed bulked samples for genetic mapping, ~30 individuals were pooled in each bulk (Liu et al, 2012). The same pooling strategy has been successfully used for numerous bulked segregant analyses. The method was cited for more than 400 times. It seems that the number of individuals in each bulk is not a concern as long as ~30 or more are used. Based on our experience, we believe 100 individuals for each bulk should account for the impact from random bias. But we agree the reviewer that a larger number may be safter for the bulked analysis. In many experiments, the number of individuals sampled may be limited by the total individuals examined. In our experiment, we started from ~2,000 calli and select typical Type I and Type II calli and avoided to select atypical ones. The choice of 100 was based on our experience and experimental results.

Reference:

Liu, Sanzhen, Cheng-Ting Yeh, Ho Man Tang, Dan Nettleton, and Patrick S. Schnable. 2012. “Gene Mapping via Bulked Segregant RNA-Seq (BSR-Seq).” PloS One 7 (5). Public Library of Science (PLoS): e36406.

Comment 5: Lines 239–241: The BSR-seq analysis found 284 SNPs associated with callus type. How many of these overlap with the GBS-identified SNPs, and could the authors quantify the degree of concordance to validate robustness?

Response 5: For BSR-seq, SNPs were directly used for analysis. For GBS, SNPs were first identified and then segment (bin) markers were developed to reduce missing data while increasing marker coverage. Therefore, we did not check the number of overlapping associated SNPs between the two mapping approaches.

Comment 6: Lines 250–256: For ctAB5a, the B73 allele is favorable for Type II callus, yet A188 is typically described as the high-regeneration line. How do the authors reconcile this apparent contradiction?

Response 6: We share the reviewer’s curiosity. Of five QTLs identified, only ctAB6a located in chromosome 6 contains the A188 allele positively contribute to form Type II callus. We think the A188 allele at the chromosome 3 locus should contribute significantly to the tissue culture response and regeneration. Based on our data, the combination of A188 alleles at the chromosome 3 locus and the chromosome 6 locus might be important for the high ability of regeneration.

Comment 7: Lines 415–417: The link between cell wall differences and nutrient/water/oxygen uptake is speculative. Do the authors have direct measurements (e.g., metabolite assays, ion flux studies) to support this, or is it entirely inferred from DEG functions?

Response 7: The speculation is based on the analyzing result from transcriptomic data. We think it is reasonable to speculate that cell wall organization may affect callus ability to uptake water, oxygen, and nutrients. Consequently, callus with different cell wall compositions may exhibit distinct growth rates and regeneration ability. Indeed, the hypothesis that wat1, the mutant reducing wall thickness, is involved in callus development is being tested in our lab. If the hypothesis is supported, the suppression of wat1 might improve the ability of regeneration.  

Comment 8: Lines 421–425: The up-regulation of MYB138 is interpreted as promoting callus proliferation, but in Arabidopsis, it reduces callus formation via enhanced GA signaling. Could the opposite correlation in maize reflect species-specific roles, or might it suggest misinterpretation?

Response 8: Thank you for the question. In the original manuscript, we did not specifically mention that myb138 was a T-DNA insertion mutant. Here is the revision with more specific information: “The MYB TF Zm00001d043131, referred to as MYB138, was previously implicated in callus induction (Ge et al. 2016). A T-DNA insertion mutant of the myb138 homologous gene reduced callus formation as compared with the wildtype in Arabidopsis, presumably through enhancing gibberellin transduction and promotes differentiation (Ge et al. 2016).”

Reference:

Ge, Fei, Xu Luo, Xing Huang, Yanling Zhang, Xiujing He, Min Liu, Haijian Lin, Huanwei Peng, Lujiang Li, Zhiming Zhang, Guangtang Pan, and Yaou Shen. 2016. “Genome-Wide Analysis of Transcription Factors Involved in Maize Embryonic Callus Formation.” Physiologia Plantarum 158 (4). Wiley: 452–62.

Comment 9: Lines 426–431: WOX and ERF transcription factors are discussed as regulators of embryogenesis. Were any of these WOX/ERF DEGs located within the mapped QTL intervals, which would strengthen their candidacy?

Response 9: Thank you for the suggestion. We did check the chromosomal locations of these genes encoding transcription factors. None of these genes are in the QTL regions.

Comment 10: Lines 448–451: The authors recommend monitoring VLCFA levels in future work. Why was lipidomic profiling not included in this study, given that transcriptomic changes strongly pointed toward fatty acid metabolism?

Response 10: This is an excellent suggestion. Our data did strongly connect the fatty acid pathway with the callus development. Lipidomic profiling and quantification would provide finer information of associations. However, for us, to start a new experiment for lipidomic profiling is not trivial. We plan to continue this project with a new graduate student. We hope to understand better if certain fatty acid components may affect the callus quality for regeneration.

Round 2

Reviewer 3 Report

Comments and Suggestions for Authors

The author has revised the MS as per the suggestions.